# Molecular Network and Culture Media Variation Reveal a Complex Metabolic Profile in *Pantoea* cf. *eucrina* D2 Associated with an Acidified Marine Sponge

**DOI:** 10.3390/ijms21176307

**Published:** 2020-08-31

**Authors:** Giovanni Andrea Vitale, Martina Sciarretta, Chiara Cassiano, Carmine Buonocore, Carmen Festa, Valerio Mazzella, Laura Núñez Pons, Maria Valeria D’Auria, Donatella de Pascale

**Affiliations:** 1Institute of Biochemistry and Cell Biology, National Research Council, 80131 Naples, Italy; giovanniandrea.vitale@ibbc.cnr.it (G.A.V.); carmine.buonocore@ibbc.cnr.it (C.B.); 2Department of Pharmacy, University of Naples “Federico II”, 80131 Naples, Italy; martina.sciarretta@unina.it (M.S.); chiara.cassiano@unina.it (C.C.); carmen.festa@unina.it (C.F.); 3Department of Integrated Marine Ecology (EMI), Stazione Zoologica Anton Dohrn, Villa Comunale, 80125 Naples, Italy; valerio.mazzella@szn.it (V.M.); laura.nunezpons@szn.it (L.N.P.); 4Department of Marine Biotechnology, Stazione Zoologica Anton Dohrn, Villa Comunale, 80125 Naples, Italy

**Keywords:** molecular networking, hydrothermal vents, *Pantoea eucrina*, biosurfactants

## Abstract

The Gram-negative *Pantoea eucrina* D2 was isolated from the marine sponge *Chondrosia reniformis*. Sponges were collected in a shallow volcanic vents system in Ischia island (South Italy), influenced by CO_2_ emissions and lowered pH. The chemical diversity of the secondary metabolites produced by this strain, under different culture conditions, was explored by a combined approach including molecular networking, pure compound isolation and NMR spectroscopy. The metabolome of *Pantoea* cf. *eucrina* D2 yielded a very complex molecular network, allowing the annotation of several metabolites, among them two biosurfactant clusters: lipoamino acids and surfactins. The production of each class of metabolites was highly dependent on the culture conditions, in particular, the production of unusual surfactins derivatives was reported for the first time from this genus; interestingly the production of these metabolites only arises by utilizing inorganic nitrogen as a sole nitrogen source. Major components of the extract obtained under standard medium culture conditions were isolated and identified as N-lipoamino acids by a combination of 1D and 2D NMR spectroscopy and HRESI-MS analysis. Assessment of the antimicrobial activity of the pure compounds towards some human pathogens, indicated a moderate activity of leucine containing N-lipoamino acids towards *Staphylococcus aureus*, *Staphylococcus epidermidis* and a clinical isolate of the emerging food pathogen *Listeria monocytogenes*.

## 1. Introduction

The Gram-negative genus *Pantoea* was only recently established as new genus within the family of Enterobacteriaceae. To date, it comprises about 20 species (https://lpsn.dsmz.de/genus/pantoea) isolated from both the host and non-host environments [1]. The high adaptability to different habitats was also accomplished by a remarkable phenotypic diversity that ranges from the pathogeny of some isolates toward plants and humans—although the actual involvement of *Pantoea* strains in the insurgence of diseases in both plants and humans is a matter of controversy—to high beneficial biotechnological and therapeutic potential.

Bioremediation of contaminated soils and water by *Pantoea* spp. was exerted through bioabsorption and metabolic transformation of heavy metal ions and/or through production of glycolipid biosurfactants, able to promote the emulsification and degradation of organic waste components and pollutants as well as of petroleum derivatives [1].

On the other hand, the mutualistic interaction with plants and insects was associated with multiple factors [2] as the metabolic cooperation, thanks to the nitrogen fixation and phosphate solubilization capabilities exhibited by some *Pantoea* species and to the production of several plant growth promoting and defense secondary metabolites [3].

Noteworthy is the production of small molecules with antimicrobial activities such pantocins, herbicolins, microcins and phenazines [1,4,5].

The study of the phenotypic features of members of this species has been so far mainly focused on the comparative genomic analysis in the search for the genetic and genomic determinants able to drive specific diversity and adaptability [6,7,8], whereas there are only sporadic papers dealing with the chemical profiling of members of this genus.

As part of our interest toward the biotechnological potential of bacterial strains isolated from unexplored and unique habitats, we conducted a collection of sponge samples in a volcanic vents system of Ischia island (Gulf of Naples, Italy). Here due to the effect of secondary volcanism, carbon dioxide seeps inflate CO_2_ in the surrounding seawater [9], creating a unique area of naturally lowered pH. Such system represents a unique in situ laboratory for studies regarding adaptations to ocean acidification (OA) on benthic biota [10]. The volcanic CO_2_ vents include areas where pH ranges from normal values (8.1–8.2) to lowered pH values (mean 7.8–7.9, minimum 7.4–7.5), mimicking the records predicted for the end of the century [11].

Marine sponges (phylum Porifera) are known to be prolific sources of bioactive metabolites, most suspected to be produced by their conspicuous microbial associates [12]. Among sponges collected in our surveys, samples from *Chondrosia reniformis* Nardo, 1847 coming from acidified zones were selected to obtain microbial isolates for downstream analyses. After culture-dependent isolations, a strain identified as *Pantoea eucrina* was selected for metabolic studies. To the best of our knowledge this is the first report of a *Pantoea* strain isolated from a marine sponge.

In this work, OSMAC (one strain many compounds) [13] and molecular networking (MN) [14] approaches were strategically combined to perform a downstream exploration of the diversity of secondary metabolites produced by *P. eucrina* D2 under different conditions. The metabolome turned out to be very complex and molecular networking analysis allowed the annotation of several metabolites that were clustered in three subgroups: lipoamino acids, diketopiperazines and surfactins.

The strain was found to be able to produce these metabolites using inorganic ammonium and nitrate salts as sole nitrogen sources. The production of specific subgroups and even of specific components within each subgroup was strongly dependent on the cultivation media, in particular the production of surfactin derivatives, here described for the first time from a *Pantoea* strain, was triggered by the use of inorganic nitrogen, while it was totally abolished under normal conditions.

The major metabolites belonging to lipoamino acids group were isolated, chemically characterized by NMR and HR-ESI-MS analysis and subjected to antimicrobial assays towards human pathogens. This hyphenate approach allowed the detection of two otherwise undetectable new surfactin derivatives, together with the isolation of several lipoamino acids derivatives, some of which never isolated before from natural sources, among them Compound **1** has never been reported even as synthetic product. Concerning the biologic activity, leucine derivatives which displayed a prominent antimicrobial activity towards Gram-positive human pathogens confirmed to be promising candidates for biotechnological applications.

## 2. Results and Discussion

### 2.1. Isolation and Identification of Bacterial Strains from the Sponge Chondrosia reniformis Living at Naturally Lowered pH Conditions

*Chondrosia reniformis*, a high microbial abundance (HMA) sponge [15] widespread in the Mediterranean Sea and with a rich microbiome [16], was collected in the marine area around Castello Aragonese, Ischia Island. This species is common in around the acidified area formed by local volcanic CO_2_ vents [17], suggesting a long term adaptation to OA conditions. The microbiome of this sponge was hence exploited as a reservoir for the isolation and culture of bacterial strains with interesting metabolite profiles. Since the sponges were collected in an acidified environment, bacterial isolation was performed in a decreasing pH gradient. Sponge pieces were brought in a sterile manner to the lab, submersed in 0.22 μm filtered seawater and plated on MB agar plates at pH: 5, 6 and 7 for the microbial isolation. No bacteria grew at pH 5, also considering agar plates frailty at that pH, while most bacteria developed on pH 6 agar plates, resulting in nine morphologically different strains, which were all identified through the 16S rRNA sequencing (Table 1).

The isolated bacteria were identified through 16S rRNA Sanger sequencing, and sequences were submitted on BLASTn (Nucleotide Basic Local Alignment Search Tool) against GenBank database for species annotation [18]. A phylogenetic tree was built including the strain D2 along with the closest matches from GenBank using MEGA X (Figure 1). These results suggest that D2 is closely related, if not conspecific, to *Pantoea eucrina*, a poorly studied member of *Pantoea* genus. This genus was established ~30 years ago in the *Enterobacteriaceae* family. It counts only for 20 isolated species [1,19,20,21], and certain strains have been found to produce antimicrobial metabolites, many of which peptides [4,5,22]. In addition, a new marine strain of *Pantoea* was recently found to produce high amounts of an exopolysaccharide [23]. This is the first record of this genus associated with a sponge, in addition, its metabolome has never been deeply investigated.

### 2.2. MS/MS-Based Molecular Networking Analysis of Pantoea eucrina D2 Metabolome Grown under Different Culture Conditions

The principle of “one strain many compounds” (OSMAC) approach is based on the principle that several microbes can potentially produce many more metabolites than they do under determined conditions. A simple and effective way to influence bacterial metabolisms is changing growth parameters such as nutrients, temperature, salinity, aeration, in this way activating or upregulating metabolic pathways which usually silenced [24]. Herein metabolites produced by the sponge-associated *P.* cf. *eucrina* D2 in different culture media were explored by modifying the sources of inorganic nitrogen and comparing them with the metabolic profile obtained in the optimal and complex TSB medium. The metabolome was investigated by analyzing the crude extracts through HPLC-MS/MS data dependent analysis. The obtained MS raw data were converted in mzXML format, processed by MZmine [25] in order to remove noise, to filter the isotopes and to align the data. Then, the output files were uploaded to Global Natural Products Social molecular networking (GNPS) [14] and used to build a feature-based molecular network [26], finally the data were exported and visualized on Cytoscape [27].

In this way, a complex web of nodes grouped in several clusters was obtained (Figure 2). Each node represents an ion found in at least one of the growth cultures, different colors are used to define their presence in different growth media, the edges size among different nodes are directly dependent on their fragmentation spectra similarity (cosine score), while the node size is directly proportional to the precursor ion intensity (sum precursor intensity). This visualization provides a scheme of the different metabolites production in the specific conditions (Figure 2).

Analysis of GNPS unique library hits and analog library hits evidenced the presence of three class of metabolites: diketopiperazines, surfactins and lipoamino acids. Herein we focused our attention on two latter classes.

#### 2.2.1. Surfactins Molecular Cluster

The analysis of the [M + H]^+^ network in Figure 3 matched on GNPS library for surfactin C (1036 *m*/*z*) and through the clusterization suggested the occurrence of related species with *m*/*z* at 1050.7041, 1064.7197, 1078.7348, 1092.7509 in addition, another pseudomolecular ion with *m*/*z* of 1106.7681 compatible with a surfactin derivative was found in the full-MS spectrum (1000–1200 Da) (Appendix A) by manual HR data curation.

Surfactins are a family of lipopeptides produced by strains of the genus *Bacillus*. They feature a cycloheptadepsipeptide core with a terminal β-hydroxy fatty acid (β-OH FA). Culture samples of different *Bacillus* spp. were found to produce complex mixtures of surfactin congeners, which differ either in the length of the fatty acid chain or in the amino acid sequence and composition of the peptide moiety.

The extract obtained in MSM C media, which contains all the detected surfactins, was further analyzed to deepen the sequence of surfactins analogs and to reveal the potential presence of isomers.

The fragmentation pattern of surfactins has been extensively investigated. The initial cleavage of the protonated ester bond was followed by sequential loss of amino acid residues giving rise to b+ and y+ series of fragment ions useful to assign the amino acid sequence, except for the discrimination of isomeric Leu and Ile and OMet Asp and Glu.

The extracted ion chromatogram (XIC) (Appendix A) obtained for all the surfactins [M + H]^+^ adducts displays for some of them more than one peak, which is ascribable to the presence of isomers, the XIC at 1036 *m*/*z* showed two peaks, the fragmentation spectrum of the first peak (Figure 4a) displayed the series of b+ fragment ions at *m*/*z* 1036(−H2O, 1018) → 923 → 808 → 695 → 596 → 483 → 370, compatible with the sequential loss of Leu/Ile7-Asp6-Leu/Ile5-Val4-Leu/Ile3-Leu/Ile2 from the C terminus. Moreover, the second typical set of y+ fragment ions at *m*/*z* 1036 (−H2O, 1018) → 667(+H2O, 685) → 554 → 441 is ascribable to the losses of C15 β-hydroxyl fatty acid chain-Glu1-Leu/Ile2-Ile/Leu3 from the precursor ion. Both series pointed toward a positional variant of surfactin C, recently reported by Khyati at al. [28], with a C15 β-hydroxyl fatty acid chain and the following amino acid sequence: Glu 1-Leu/Ile 2-Leu3-Val4-Leu/Ile5-Asp6-Leu/Ile7.

The second isoform with *m*/*z* 1036.7 (Figure 4b) gave a series of b+ ions at *m*/*z* 1036 (–H2O, 1018) → 937 → 808 → 709 → 596 → 384, compatible with the loss of Val7–OMeAsp or Glu6-Val5-Leu/Ile4-Leu/Ile3-Val2. y+ fragment ions at *m*/*z* 1036 (−H2O, 1018) → 653 (+H2O, 671) → 554 → 441 are also in line with proposed sequence and with a C16 β-hydroxyl fatty acid chain.

Although the difference of 129 between b6 and b5 ions could be ascribable both to a Glu residue or to the presence of an OMet-Asp, we proposed here the C16 β-hydroxyl fatty acid chain-Glu1-Val2-Leu/Ile3-Leu/Ile4-Val5-OMeAsp6-Val7 structure based on the observation of a ion fragment a 905, likely arising from b6 ion by MeOH loss. To the best of our knowledge, this surfactin isoform is described in this study for the first time.

Similarly, two fragmentation patterns were observed from precursor ions at 1050.7. The first one (Figure 4c) could be assigned with high confidence to C15 β-hydroxyl fatty acid chain-Glu1-Leu/Ile2-Leu/Ile3-Leu/Ile4-Leu/Ile5-Asp6-Leu/Ile7 on the basis of b+ and y+ series (b+: 937; 822; 709; 596; 483; 370—y+: 681 (+H2O, 699); 568; 455; 342) [28].

The second isomer at *m*/*z* 1050.7 (Figure 4d) showed a fragmentation pattern compatible with the substitution of Leu/Ile in position 4 with a Val and a fatty acid chain with 16 carbons (C16 β-hydroxyl fatty acid chain-Glu1-Leu/Ile2-Leu/Ile3-Val4-Leu/Ile5-Asp6-Leu/Ile7 b+: 937; 822; 709; 610; 497; 384—y+: 667 (+H2O, 685); 554; 441) [28]. Based on fragmentation profile, the three species at *m*/*z* 1064.7 1078.7, 1092.7, (Figure 5a–c) could be assigned as C16, C17 and C18 β-hydroxyl fatty acid homologs with the same peptide moiety, the C18 β-hydroxyl fatty acid isoform is here described for the first time [28].

Finally, two homologs with Glu1-Leu/Ile 2-Leu/Ile3-Leu/Ile4–OMeAsp or Glu5-Leu/Ile6-Leu/Ile7 as aminoacidic sequence and a C17 and C18 β-hydroxyl fatty acid chain were found to be ascribable to the ions with *m*/*z* at 1092.7 and 1106.7 (Figure 5d,e) [29,30].

A peculiar feature of the surfactin analogs produced by *P.* cf. *eucrina* D2 seems to be the presence of the four consecutive leucine motifs found in certain derivatives, and the reversed position of Leu5 and Asp6 compared to the canonical surfactin C (Appendix A). The production of surfactins with C17/C18 β-hydroxyl fatty acid chain appear a further unusual feature, with few literature precedents [30].

To the best of our knowledge, this is the first report of the production of surfactins from a Pantoea strain. This is not surprisingly since the production of surfactins was not observed using conventional growth conditions (i.e., TBS medium) whereas it was triggered in the OSMAC media using inorganic sources of nitrogen NH_4_Cl or NaNO_3_. Similarly, the use of inorganic nitrogen sources demonstrated to be an effective tool to simulate the production of the antimicrobial peptide SBR-22 in the marine *Streptomyces psammoticus* BT-408 strain [31] and of the pigment pyocyanin in *Pseudomonas* sp. MCCB-103 [32]. As concerning the carbon sources, glycerol was found to enhance the production, when compared to glucose. The use of glycerol as cost-effective substrate for biosurfactants production by wild type and engineered strains of *Pseudomonas* has been extensively explored and rationalized [33], whereas the use of glycerol in surfactin production has been only sporadically explored [34]. Unfortunately, surfactins derivatives were detected as minor components co-eluting with major lipoamino acids, therefore we were unable to perform an isolation and chemical characterization work.

#### 2.2.2. Lipoamino Acids Molecular Cluster

Figure 6 reports the second cluster featuring a huge number of nodes. The node at *m*/*z* 404.3149 matched the 2-(14-methylpentadecanoylamino)-3-phenylpropanoic acid, a member of lipoamino acids (also referred as N-acyl amino acids), namely conjugates of one amino acid unit linked via an amide bond to saturated or unsaturated fatty acids. In particular, this compound was recently reported as component of a complex mixture of related lipoamino acids from an entomopathogenic *Pantoea* sp. strain isolated from an individual insect from the *Diaspididae* family [24]. This suggested that the members of this cluster could represents lipoamino acid variants differing for the amino acid and/or fatty acid moiety.

Comprehensive manual dereplication of each node within the cluster was done by determination of the molecular formula, from the analysis of the positive pseudomolecular ion [M + H]^+^, and from inspection of the fragmentation pattern. The key fragment arising from the cleavage of the amide bond allowed to assign the amino acid portion, and by subtraction, the length of the fatty acid subunit, whereas the presence of double bonds was inferred from the calculated round double bond equivalent (RDBE). Through the above analysis, the annotation of each node within the cluster was done and putative phenylalanine, leucine/isoleucine and valine derivatives were found with fatty acid chains ranging from C10 to C21, some of which containing one or two insaturations or an hydroxyl group, the detailed assignments are summarized in Appendix A.

Our results parallel those reported by Tourè et al. [8] in what concerns the qualitative composition of the mixture except for some congeners. Although our *Pantoea* cf. *eucrina* D2 was found to produce the phenylalanine conjugates as minor components, whereas in all tested media cultivation conditions the Leu/Ile conjugates are the major components.

Interestingly, we observed that *P.* cf. *eucrina* D2 was able to produce lipoamino acids in all tested growing media containing only inorganic nitrogen sources, so showing nitrifying properties.

### 2.3. Isolation and Structure Elucidation of Pure Compounds

In order to solve the Leu/Ile and the configurational ambiguities left by the MS analysis and to assess the antimicrobial activity of the major pure components, a medium scale cultivation (1.8 L) in TBS medium was performed, the crude extracellular extract (1.0 g) was fractionated by reverse Phase MPLC, and the obtained fractions were purified on UPLC equipped with a semipreparative PFP column as described in Material and methods section, affording pure Compounds **1**–**8** (Figure 7).

The structures of these compounds were determined by combined MS/MS and 2D NMR analysis and by comparison with literature data.

Compound **1** gave a pseudomolecular ion [M + H]^+^ at *m*/*z* 412.3411 [M + H]^+^, calcd. 412.3411 Δ = −3.84 ppm corresponding to the molecular formula C_24_H_45_NO_4_. LC-HRMS/MS fragmentation showed a diagnostic molecular ion at *m*/*z* of 132.1017 corresponding to C_6_H_14_O_2_ N (Ile and/or Leu).

The analysis of 2D COSY, HSQC and HMBC allowed the straightforward assignment of the leucine aminoacyl spin system (Appendix A). Particularly diagnostic for the Ile/leu discrimination are the ^1^H and ^13^C NMR chemical shifts of two γ methyl groups (δ_H_ 0.96 and 0.97/δ_C_ 21.8 and 23.2).

Remaining NMR data were indicative of a linear acyl group (one terminal methyl δ_H_ 0.90, t, *J* = 7.0 Hz) with a double bond (δ_H_ 5.36, δ_C_ 130.1) and one hydroxy group (δ_H_ 4.00, δc 68.9). Key COSY and HMBC correlations were reported in Figure 8. COSY correlations between the diastereotopic methylene protons at C-2′ (δ_H_ 2.50 (d, *J* = 14.7 Hz) and 2.34 (dd; *J* = 14.7, 8.9 Hz) and the hydroxymethine H-3 (δ_H_ 4.00) and between H-3 and methylene protons H_2_-4′ (δ_H_ 1.56 and 1.48) together the diagnostic HMBC correlations between H_2_-2/C-1′, H_2_-2/C-3′, H_2_-2/C-4′ and H_2_-4/C-3 clearly suggested the presence of 3-hydroxy unsaturated fatty acid moiety. The mass and NMR data allowed us to determine that the fatty acid moiety is 3-hydroxyoctadec-9-enoic acid.

The stereochemistry of the double bond was assigned as *Z*, based on the ^13^C chemical shift of the C8-C11 allylic carbons at δ_C_ 27.4 ppm [35].

Marfey’s analysis confirmed the identity of Leu residue and its L-configuration (Appendix A). At this stage, the stereochemistry of hydroxy-methine in position 3 remains undefined. Compound **1** was therefore identified as (Z)-*N*-(3-hydroxyoctadec-9-enoyl)-l-leucine.

Structural *N*-acylamino acid analogs **2–5** were characterized using comparative NMR, HRMS/MS and Marfey’s analysis. Compounds **2–4** were already isolated as mixture of enantiomers and an inseparable mixture together with their isoleucine congeners from *Cobetia* sp. isolated from the marine hydroid *Hydractinia echinata* [36].

*N*-palmitoyl-l-leucine (**4**) was already isolated by *Streptomyces* sp. and identified as a late stage inhibitor of m-RNA splicing through a high- throughput screening of 5304 pre-fractions of marine bacterial lysates [37].

Through NMR and MS analysis and by comparison with literature data [8] Compounds **6**, **7** and **8** were identified as *N*-palmitoyl-l-phenylalanine, *N*-palmitenoyl-l-phenylalanine and *N*-oleyl-l-phenylalanine, respectively. *N*-palmitenoyl-l-phenylalanine (**7**) was already isolated from *Pantoea* sp., whereas Compounds **6** and **8** were obtained in the same study by synthesis [8]. The configuration of the double bond in the acyl chain of **7** and **8** was assigned as Z on the basis of the ^13^C NMR chemical shifts of allylic methylenes, whereas the L-configuration of the phenylalanine residue was assigned by comparison of the experimental positive optical rotation data with those reported in the literature [8].

### 2.4. Minimum Inhibition Concentration (MIC)

The data regarding antimicrobial activity of lipoamino acids are often dated and not widely described, in the specific case, are not reported at all for leucine N-acylamino lipids. Herein the antimicrobial potential of the pure molecules was determined by microdilution method and the MIC value was reported for each pathogen, DMSO at an initial concentration of 2% (*v*/*v*) was adopted as negative control and antibiotics at their respective breakpoint concentrations were used as positive controls. The results reported in Table 2 show that this class of molecules is ineffective against the Gram-negative strains (*Acinetobacter baumannii* 13 and *Stenotrophomonas maltophilia* ATCC 13637). In addition, the amino acid moiety seems to play an important role in the exertion of antimicrobial properties, in fact, in accordance with a recent report on phenylalanine lipoamino acids [8], Compounds **6**–**8** are totally ineffective or slightly active towards *Staphylococcus aureus* and also against *Staphylococcus epidermidis*, independently from fatty acid chain length. On the other hand, leucine- containing congeners were very active towards *Staphylococcus* spp., with the best MIC values given by Compounds **1**, **3** and **4** with 10 (μg/mL) towards *S. aureus* and by Compounds **3** and **4** with 10 (μg/mL) towards *S. epidermidis*.

The activity of these biosurfactants towards foodborne pathogens was observed on certain shorter chain derivatives [38], where *N*-myristoyl-l-phenylalanine showed a MIC of 34 (μg/mL) towards *L. monocytogenes*, although longer chain congeners herein isolated seems to be more effective towards the clinical isolated *L. monocytogenes* strain used in this study. Additionally, it is noticeable that phenylalanine congeners hold a comparable activity with leucine one, with the best activity value obtained for Compounds **3** and **6** (4.0 μg/mL).

## 3. Materials and Methods

### 3.1. General Experimental Procedures

1D and 2D NMR experiments were recorded on Bruker Avance NEO 400 MHz and 700 MHz spectrometers (Bruker, USA) with a RT-DR-BF/1H-5mm-OZ SmartProbe. Chemical shifts were reported in δ (ppm) and were referenced to the residual CDCl_3_ as internal standards (δ_H_ = 7.26 and δ_C_ = 77.0 ppm) and CD_3_OD as internal standards (δ_H_ = 3.31 e δ_C_ = 49.0 ppm). All of the recorded signals were in accordance with the proposed structures. Spin multiplicities are given as s (singlet), br s (broad singlet), d (doublet), t (triplet) or m (multiplet).

The LC-HRMS and MS/MS analysis were carried out on an LTQ XL-Orbitrap high-resolution mass spectrometry system (Thermo Scientific) equipped with an Accelera 600 pump HPLC (LCHRMS).

Further fragmentation analysis was carried out on an LTQ XL mass spectrometry system (Thermo Scientific) equipped with a HESI source and connected to an Ultimate 3000 HPLC pump.

The first purification step was run on a User Manual PuriFlash XS 520 Plus equipped with UV detector. Purification of single molecules was performed on an Acquity UPLC H-CLASS connected to a PDA detector (Waters).

The 96-well plates were read on a Biotek ELX800, monitoring the absorbance at 600 nm at room temperature.

### 3.2. Media and Buffers

Tryptic soy broth (TSB): 3.0-g/L papaic digest of soya, 2.5-g/L D (+)-glucose, 17-g/L pancreatic digest of casein, 2.5-g/L di-potassium hydrogen phosphate, 5-g/L sodium chloride.

Modified mineral salt medium (MSM mod): 0.7-g/L KH_2_PO_4_, 0.9-g/L Na_2_HPO_4_, 0.4-g/L MgSO_4_, 0.1-g/L CaCl_2_.

MSM A: MSM mod + glucose 1% *v*/*v* + 1-g/L NH_4_Cl

MSM B: MSM mod + glucose 1% *v*/*v* + 2-g/L NH_4_Cl

MSM C: MSM mod + glycerol 1% *v*/*v* + 1-g/L NH_4_Cl

MSM D: MSM mod + glucose 1% *v*/*v* + 2-g/L NaNO_3_

MSM E: MSM mod + glucose 1% *v*/*v* + 4-g/L NaNO_3_

Cation-adjusted Mueller–Hinton broth (CAMHB) [39]

Marine broth (MB): 19.4-g/L NaCl, 8.8-g/L MgCl_2_, 5-g/L peptone, 3.24-g/L Na_2_SO_4_, 1.8-g/L CaCl_2_, 1-g/L yeast extract, 0.55-g/L KCl, 0.16-g/L NaHCO_3_, 0.10-g/L Fe(III) citrate, 0.08-g/L KBr, 0.034-g/L SrCl_2_, 0.022-g/L H_3_BO_3_, 0.008-g/L Na_2_HPO_4_, 0.004-g/L sodium silicate, 0.0024-g/L NaF, 0.0016-g/L NH_4_NO_3_

### 3.3. Sponge Collection and Bacterial Isolation at Different pH

Specimens of the sponge *Chondrosia reniformis* were collected by scuba diving at 2-m depth in several spots influenced by CO_2_ emissions and low pH close to Castello Aragonese, Ischia, Italy (40°43.84′ N 13°57.08′ E). Immediately after the collection some sponge pieces were stored at 15 °C submersed in 0.22 μm filtered seawater and taken to the laboratory for the microbial isolation. Bacterial isolation was performed at different pH values. Sponge pieces were shredded and mixed with 20 mL sterile seawater, then the solution was shaken for 30 min, and finally the bacterial suspension was serially diluted (10^−1^, 10^−2^ and 10^−3^ in 10 mL) and 100 μL of each dilution was plated on different MB agar plates, previously adjusted with HCL 1 M to the three chosen pH values: 5, 6 and 7.

The petri dishes were incubated at 15 °C for 15 days, which is the same temperature of the seawater registered during the sampling session. After the incubation period, morphologically different colonies were picked by a sterile loop, grown in liquid MB broth and finally pure bacteria cultures were stored at -80 °C in sterile cryovials with glycerol 20% *w*/*w*.

### 3.4. 16S Sequencing

PCR was carried out in a total volume of 50 µL containing 25 µL of PCR Master Mix 2× (a ready-to-use solution containing TaqPol, buffer, MgCl2 and dNTPs), 0.2 µM of both primers 27F (5′-AGAGTTTGATCCTGGCTCAG-3′) and 1492R (5′-GGTTACCTTGTTACGACTT-3′), and 100 ng of DNA. PCR protocol is reported in Appendix A. Five microliters of each PCR product were run on 1% agarose gel at 110 V for 45 min to check the quality of DNA and observed under UV light. Then PCR products were purified with the GeneAll kit according to manufacturer’s instructions and the obtained amplicons were send to Eurofins Genomics for the sequencing. Finally, the sequences were used as template for a taxonomic annotation via BLASTn tool against GenBank database, using the 16S RNA sequences collection. The contig obtained submitting the forward and the reverse sequences to Prabi CAP3 (http://doua.prabi.fr/software/cap3) was submitted to BLASTn for the affiliation analysis. Evolutionary analyses involved ten nucleotide sequences, our D2 strain along with nine close matching entries according to BLASTn, and were conducted in MEGA X [40]. A phylogenetic tree was inferred using the neighbor-joining method [41]. The evolutionary distances were computed using the Kimura 2-parameter method [42] and are in the units of the number of base substitutions per site. All ambiguous positions were removed for each sequence pair (pairwise deletion option). Bootstrap values were calculated with 1000 resamples.

### 3.5. Bacterial Cultivation, OSMAC Cultures and Extraction Methodologies

To produce the extracts in the complex media TSB, a single CFU of D2 was used to inoculate 3 mL of TSB, in sterile bacteriological tubes. After 24 h of incubation at 20 °C and 210 rpm, the inoculum was employed to inoculate 125 mL of the respective media in a 500-mL flask, at the initial cell concentration of 0.01 OD_600_/mL. The flask was incubated for 5 days at the same conditions.

The OSMAC (one strain many compounds) approach was used to trigger the production of metabolites unexpressed or underexpressed under normal conditions. Different D2 cultures have been set utilizing MSM mod liquid media as base, glucose or glycerol as carbon source, while the inorganic nitrogen salts NH_4_Cl or NaNO_3_ were used as sole nitrogen sources at different concentrations. In this case, a different protocol was used to inoculate the strain: a D2 single colony was picked from a pure plate and inoculated in 3 mL of TSB. After 2 days of incubation at 20 °C and 210 rpm, the culture was centrifuged, the medium was discarded, and the cells were resuspended in 1 mL of MSM mod. The absorbance was measured at the spectrophotometer and was inoculated in 125 mL of each different medium in a 500-mL flask at the same conditions previously described.

The medium-scale fermentation was set by inoculating 1.8 L of TSB with the same procedure described above at 20 °C for 5 days to afford 1 g of crude extract.

After the incubation time, all the cultures were centrifuged at 6800× *g* at 4 °C for 45 min, and the supernatant was extracted with 2 volumes of ethyl acetate, then the organic phase was dried under vacuum at the rotary evaporator to afford the crude extract.

### 3.6. Mass Spectrometry Analysis

The TSB and OSMAC extracts were first subjected to a desalination step by using the Sep-Pak tC18 Plus Short Cartridge 400 mg. Each extract was dissolved in the minimum amount of MeOH and upload on a cartridge, then 3 beds of MQ H_2_O were used to wash the extract and 3 beds of MeOH and MeOH + TFA 0.01% to eluate the metabolites. Then the methanolic extracts were dried under nitrogen flux and subjected to mass spectrometric analysis.

Both low resolution and high resolution data dependent analysis were carried out performing chromatographic separation of samples on a Synergi 2.5 mm Fusion_RP 100 × 2 by means of a linear gradient of B in 27 min (Buffer A: H_2_O + 0.1% Formic acid (FA), Buffer B: ACN 0.1% + FA). Full MS spectra were acquired in the mass range 150–1500 with 10 data-dependent MS/MS events of the 10 most intense ions.

### 3.7. Molecular Networking Building

The MS data were treated with MZmine [25] upon being converted from *.raw extension to *.mzXML extension using the tool MSConvert by ProteoWizard.

Version 2.53 of MZmine was used to process the data and the parameters used in each step are listed in the Appendix A, the job was exported as two output data, one quantification table (.csv) and one file containing the MS and MS/MS features (.mgf).

These data were submitted to GNPS website (https://gnps.ucsd.edu) [14] and the network was created with the feature-based molecular networking (FBMN) workflow [26]. The parameters were changed in accordance with the data and the used mass spectrometer, the precursor ion mass tolerance was set to 0.05 Da, the fragment Ion mass tolerance was set to 0.05 Da. The molecular network was created with a maximum of 100 nodes for each cluster, edges were filtered to have cosine score above 0.7 and at least 3 matching peaks, more edges among two nodes were kept only if each of the nodes is, respectively in each other list of the 10 more similar nodes. Analogs for each node were also searched, with a maximum mass difference of 300 Da, here at the same way all the analogs hits had to show a cosine score above 0.7 and at least 3 matching peaks. Finally the network was visualized by Cytoscape [27].

The MN job can be publicly accessed through this link https://gnps.ucsd.edu/ProteoSAFe/status.jsp?task=26db58f686234915b9e8884718f25085.

### 3.8. Medium-Scale Cultivation and Isolation of Pure Lipoamino Acids

The medium-scale fermentation was set by inoculating 1.8 L of TSB with the same procedure previously described for the small cultures, at 20 °C for 5 days.

The culture was centrifuged at 6800× *g* at 4 °C for 45 min, and the supernatant was extracted with 2 volumes of ethyl acetate, then the organic phase was dried under vacuum at the rotary evaporator to afford about 1 g of crude extract. The extracellular extract was fractionated in two runs by MPLC on RP-column (C18 spherical 20–35-µM 100A, 12 g) using a linear gradient of H_2_O/MeOH (*v*/*v*, from 90:10 to 0:100 over 1 h) to give six main fractions, labeled A to G. Fraction F (110 mg), eluted with MeOH:H_2_O 90:10, was purified by UPLC using a Luna^®^ PFP(2) column (5 µM, 250 mm × 10 mm i.d.) with ACN/H_2_O 62% with 0.1% TFA as mobile phase (flow rate 2.00 mL/min) giving Compounds **1** (1.6 mg, t_R_ 32 min); **2** (5 mg, t_R_ 34 min), **4** (11 mg, t_R_ 39 min) and **7** (0.6, t_R_ 41 min). Fraction G (160 mg), eluted with MeOH:H_2_O 90:10, was subjected to UPLC using a Luna^®^ PFP(2) column (5 µM, 250 mm × 10 mm i.d.) and ACN/H_2_O 62% with 0.1% TFA as mobile phase (flow rate 2.00 mL/min) to yield Compounds **3** (9.9 mg, t_R_ 42 min), **6** (3.3 mg, t_R_ 44 min), **5** (8 mg, t_R_ 47 min) and **8** (7 mg, t_R_ 49 min).

Compound **1**: white powder; [α]_D_^25^ -7 (c 0.16, MeOH); ^1^H and ^13^C NMR (700 and 175 MHz, CDCl_3_) spectroscopic data, see Appendix A; HR-ESIMS *m*/*z* 412.3411 [M + H]^+^ (calculated for C_24_H_45_O_4_ N^+^, 412.3421).

*N*-myristoyl-l-leucine (**2**): white powder; [α]_D_^25^ − 9 (c 0.5, MeOH); ^1^H NMR (400 MHz, CDCl_3_) spectroscopic data, see Appendix A, δ_H_ 6.06 (NH, d, *J* = 6.9 Hz), 4.58 (1H, m), 2.26 (2H, t, *J* = 7.4 Hz), 1.72 (2H, m), 1.65 (3H, m), 1.29 (20H, m), 0.98 (3H, d, *J* = 6.2 Hz), 0.96 (3H, d, *J* = 6.2 Hz), 0.90 (3H, t, *J* = 6.8 Hz); HR-ESIMS *m*/*z* 342.2995 [M + H]^+^ (calculated for C_20_H_40_O_3_N^+^, 342.3003).

*N*-palmitoyl-l-leucine (**3**): white powder; [α]_D_^25^ − 14 (c 0.34, MeOH); ^1^H and ^13^C NMR (700 and 175 MHz, CDCl_3_) spectroscopic data, see Appendix A; HR-ESIMS *m*/*z* 370.3308 [M + H]^+^ (calculated for C_22_H_44_O_3_N^+^, 370.3316).

*N*-palmitoleoyl-l-leucine (**4**): white powder; [α]_D_^25^ − 7 (c 1.0, MeOH); ^1^H and ^13^C NMR (400 and 100 MHz, CDCl_3_) spectroscopic data, see Appendix A; HR-ESIMS *m*/*z* 368.3150 [M + H]^+^ (calculated for C_22_H_42_O_3_N^+^, 368.3159).

*N*-oleoyl-l-leucine (**5**): white powder; [α]_D_^25^ − 8 (c 0.41, MeOH); ^1^H and ^13^C NMR (400 and 100 MHz, CDCl_3_) spectroscopic data, see Appendix A; HR-ESIMS *m*/*z* 396.3463 [M + H]^+^ (calculated for C_24_H_46_O_3_N^+^, 396.3472)

*N*-palmitoyl-l-phenylalanine (**6**): white powder; [α]_D_^25^ + 41 (c 0.04, MeOH); selected ^1^H NMR (400 MHz, CD_3_OD) spectroscopic data, see Appendix A; δ_H_ 7.25 (5H, m), 4.68 (1H, dd, *J* = 9.5 and 4.5 Hz), 3.22 (1H, dd, *J* = 14.0 and 4.5 Hz), 2.94 (1H, dd, *J* = 14.0 and 9.5 Hz), 2.16 (2H, t, *J* = 7.4 Hz), 1.49 (2H, m), 1.29 (24H, m), 0.90 (3H, t, *J* = 6.8 Hz); HR-ESIMS *m*/*z* 404.3149 [M + H]^+^ (calculated for C_25_H_42_O_3_N^+^, 404.3159).

*N*-palmitoleoyl-l-phenylalanine (**7**): white powder; [α]_D_^25^ + 20 (c 0.06, MeOH); (400 MHz, CD_3_OD) spectroscopic data, see Appendix A; δ_H_ 7.25 (5H, m), 5.35 (2H, m), 4.64 (1H, dd, *J* = 9.6 and 4.8 Hz), 3.24 (1H, dd, *J* = 13.9 and 4.7 Hz), 2.93 (1H, dd, *J* = 13.9 and 9.3 Hz), 2.14 (2H, t, *J* = 7.4 Hz), 2.04 (2H, m), 1.49 (2H, m), 1.31 (16H, m), 0.89 (3H, t, *J* = 7.0 Hz); HR-ESIMS *m*/*z* 402.2993 [M + H]^+^ (calculated for C_25_H_40_O_3_N^+^, 402.3003).

*N*-oleoyl-l-phenylalanine (**8**): white powder; [α]_D_^25^ + 44 (c 0.06, MeOH); ^1^H NMR (400 MHz, CD_3_OD) see Appendix A; δ_H_ 7.24 (5H, m), 5.36 (2H, m), 4.68 (1H, dd, *J* = 9.6 and 4.8 Hz), 3.21 (1H, dd, *J* = 13.9 and 4.8 Hz), 2.94 (1H, dd, *J* = 13.9 and 9.6 Hz), 2.13 (2H, t, *J* = 7.5 Hz), 2.04 (2H, m), 1.48 (2H, m), 1.30 (20H, m), 0.90 (3H, t, *J* = 7.0 Hz); ^13^C NMR (100 MHz, CD_3_OD) see Appendix A, δ_C_ 176.2, 175.7, 138.9, 131.0, 130.6 (2C), 129.5, 127.7, 55.5, 38.6, 37.3, 33.2, 31.1 (2C), 30.9 (2C), 30.8 (2C), 30.7 (2C), 30.5, 30.4, 28.5 (2C), 27.3, 23.9, 14.6; HR-ESIMS *m*/*z* 430.3304 [M + H]^+^ (calculated for C_27_H_44_O_4_N^+^, 430.3316).

### 3.9. Hydrolysis and Advanced Marfey’s Analysis.

Compound **3** (0.8 mg) was hydrolyzed with 6-N HCl 120 °C for 12 h. The residual HCl fumes were removed under vacuum. The hydrolysate of **3** was split and dissolved in TEA/acetone (2:3, 100 μL), and the solution was treated with 100 μL of 1% 1-fluoro-2,4-dinitrophenyl-5-d-alaninamide (d-FDAA) and 1-fluoro-2,4-dinitrophenyl-5-l-alaninamide (l-FDAA) in CH_3_CN/acetone (1:2).

The vial was heated at 50 °C for 1.5 h. The mixture was dried, and the resulting d-FDAA and l-FDAA derivatives were dissolved in MeOH (200 μL) for subsequent analysis. Authentic standards of l-Leu, l-Ile and d-Allo-Ile were treated with l-FDAA and d-FDAA as described above and yielded the l-FDAA and d-FDAA standards. Marfey’s derivatives of **3** were analyzed by LC–ESI–MS, and their retention times were compared with those from the authentic standard derivatives. A Luna Omega 3 μm Polar C18 column (100 × 2.1 mm) maintained at 25 °C was eluted at 300 μL/min with 0.1% HCOOH in H_2_O and ACN. The gradient program was as follows: 10% ACN 2 min, 10% → 95% ACN over 10 min, 100% ACN 3 min. Mass spectra were acquired in positive ion detection mode, and the data were analyzed using the Xcalibur suite of programs.

### 3.10. Minimum Inhibition Concentration (MIC) Assessment

The antimicrobial activity of the pure lipoamino acids was determined by microdilution method and MIC values were determined comparing them with appropriate antibiotics, how described by the Clinical and Laboratory Standard Institute (CLSI) [39]. The antimicrobial assay was performed in CAMHB as medium, pure compounds were dissolved in DMSO and were 2-fold serially diluted from in a final volume of 100 μL of CAMHB medium in a 96-well microtiter plate (Sarstedt), obtaining concentration in the range 128–1 μg/mL. DMSO at an initial concentration of 2% (*v*/*v*) was adopted as negative control. Each well contained 50 μL of a pure molecule solution at twice the desired final concentration, therefore it was inoculated with 50 μL of bacterial culture grown overnight at 37 °C, resulting in a final inoculum of 4 × 105 CFU/mL in a 100 μL final volume of each well. Then, each plate was incubated for 20 h at 37 °C to allow optimal bacterial growth. The pathogenic strains used in the liquid inhibition assay are: *S. aureus* 6538P [43], *L. monocyogenes* MB677 [44], *S. epidermidis* ATCC 35,984 [45], *A. baumannii* 13 [46] and, *S. maltophilia* [47].

## 4. Conclusions

In this study, MN and OSMAC integrated strategies were used to study the secondary metabolism of the endophytic strain *Pantoea* cf. *eucrina* D2 isolated from the sponge *Chondrosia reniformis* adapted to OA conditions. The study evidenced the production of otherwise silent surfactin biosurfactants induced by using inorganic nitrogen as sole nitrogen source.

The molecular networking-based approach confirmed the production of lipoamino acids as chemotaxonomic markers of members of *Pantoea* genus [8] and guided towards the isolation of six new structural variants of this growing family of bacterial secondary metabolites, together with two known derivatives.

All isolated compounds exhibited antimicrobial activity. It is currently suggested that lipoamino acids exert a protective role in the endosymbiotic interaction with the host [8] and that the observed species-specific tight regulation of the composition of lipoamino acid mixtures [48,49] may be the result of the epigenetic regulation of their biosynthesis in response to specific environmental pressures. The leucine-based N-acylamino acids were found to possess a good antimicrobial potential towards Gram-positive strains, in particular towards a clinical isolate of the foodborne pathogen *L. monocytogenes*. The natural presence of *Pantoea* cf. *eucrina* D2 in *Chondrosia reniformis* may likely afford the sponge with advantages, in what regards defense from other undesirable invasive microbes by out-competition and antimicrobial properties. However, also, the production of exopolysaccharides and surfactant-type chemicals, may create protective envelopes for beneficial endosymbiotic microorganisms (e.g., N-fixing microbes) sensitive to low pH scenarios [1,23].

Further studies will be devoted to the optimization of the production of both biosurfactant families and to the exploration of the biotechnological potential of *Pantoea eucrina* D2 in bioremediation, biocontrol and food preservation fields.

## Figures and Tables

**Figure 1 ijms-21-06307-f001:**
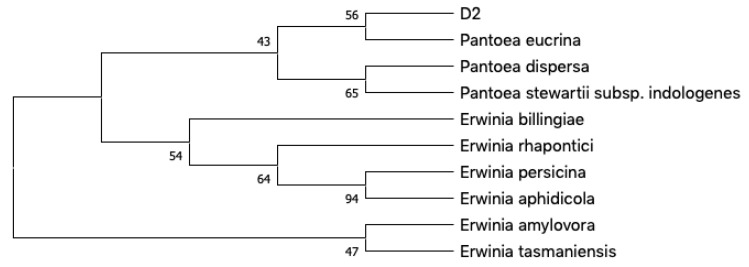
Phylogenetic neighbor-joining (NJ) tree generated with MEGAX based on 16S rRNA gene sequences from D2 and the most related species according to matching entries on BLAST. Next to the branches are shown the percentage of replicate trees in which the associated taxa clustered together in the bootstrap test (1000 resamples).

**Figure 2 ijms-21-06307-f002:**
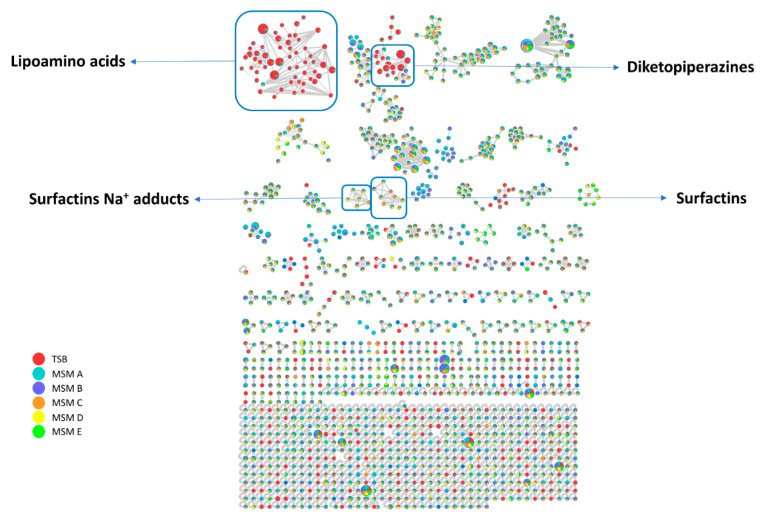
*Pantoea* cf. *eucrina* D2 metabolic network overview, it was obtained through global natural products social molecular networking (GNPS) and Cytoscape. The presence of lipoamino acids, diketopiperazines and surfactins clusters are displayed.

**Figure 3 ijms-21-06307-f003:**
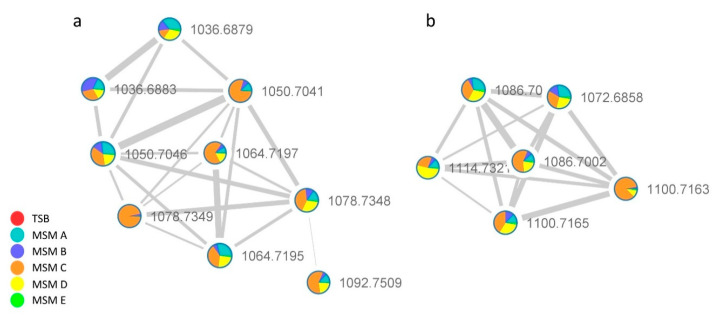
Surfactins (**a**) [M + H]^+^ and (**b**) [M + Na]^+^ adduct clusters. Color chart shows that they are only produced in the media containing inorganic nitrogen sources.

**Figure 4 ijms-21-06307-f004:**
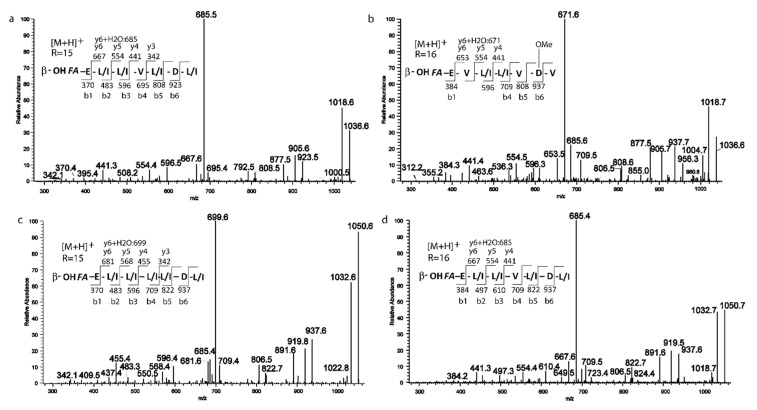
MS/MS spectra of the two isomers for the precursors (**a**,**b**) 1036.6 *m*/*z*; (**c**,**d**) 1050.6 *m*/*z*.

**Figure 5 ijms-21-06307-f005:**
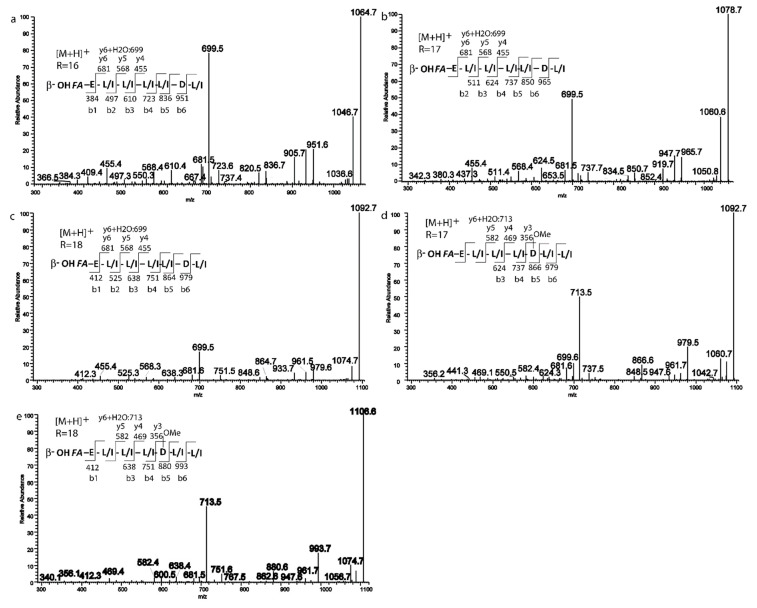
MS/MS spectra of precursors at (**a**) 1064.7 *m*/*z*; (**b**) 1078.7 *m*/*z*; (**c**) 1092.7 *m*/*z*; (**d**) 1092.7 *m*/*z*; (**e**) 1106.6 *m*/*z*.

**Figure 6 ijms-21-06307-f006:**
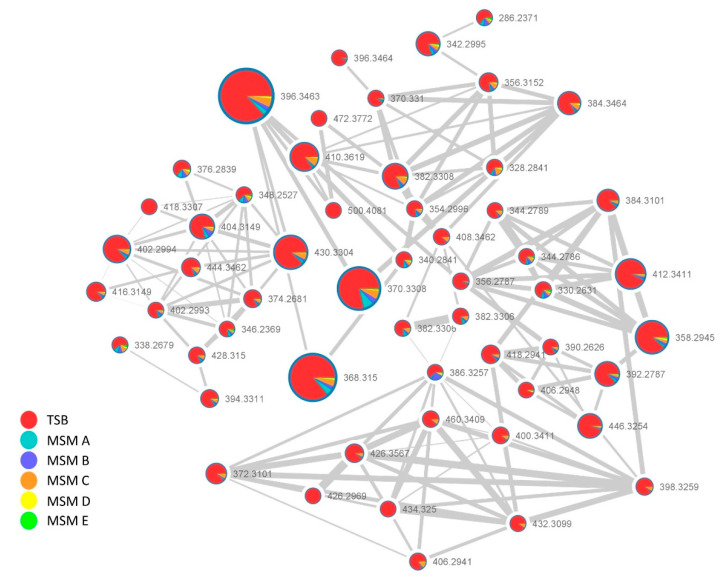
Lipoamino acids cluster in *P.* cf. *eucrina* D2, the size of the nodes is directly proportional to the precursor ion intensity, while the color (explained in the color chart) is dependent on the growth media in which the ion was detected.

**Figure 7 ijms-21-06307-f007:**
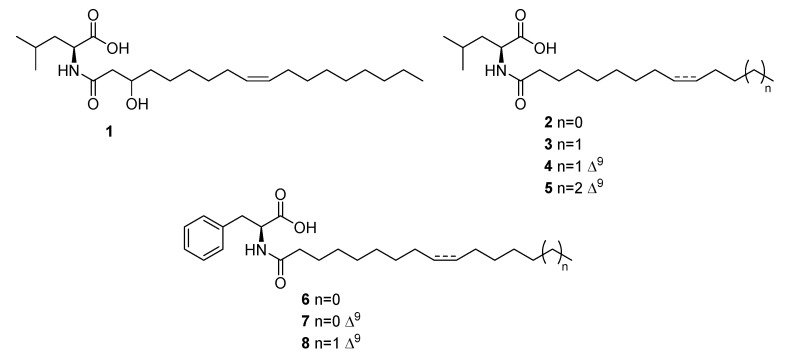
Structures of isolated aminolipids **1–8**.

**Figure 8 ijms-21-06307-f008:**
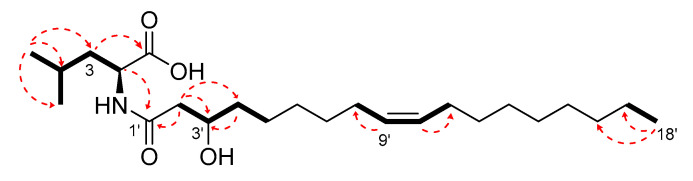
Key COSY (bold lines) and HMBC (red dashed arrows) correlations of Compound **1.**

**Table 1 ijms-21-06307-t001:** Genera of the bacterial isolates at pH 6 based on their 16S sequences.

Strain	Identity
**G**	*Pseudoalteromonas* sp.
**H**	*Shewanella* sp.
**S1**	*Pseudomonas* sp.
**U1**	*Sphingobium* sp.
**Z1**	*Sphingobium* sp.
**X1**	*Stenotrophomonas* sp.
**J1**	*Acinetobacter* sp.
**D2**	*Pantoea* sp.

**Table 2 ijms-21-06307-t002:** Antimicrobial activity expressed as MIC values for the pure Compounds 1–8 towards a panel of human pathogens.

MIC (μg/mL)					
Compound	*S. aureus* 6538P	*L. monocyogenes* MB677	*S. epidermidis* ATCC 35984	*A. baumannii* 13	*S. maltophilia* ATCC 13,637
1	10	8.0	13	–	–
2	13	8.0	13	–	–
3	10	4.0	10	–	–
4	10	6.6	10	–	–
5	128	6.6	128	–	–
6	–	4.0	–	–	–
7	64	16	128	–	–
8	–	26	–	–	–
Positive control ^a^	2	1	1.7	4	3.3

Each experiment was repeated at least three times and the mean value is reported. ^a^ Vancomycin was used as positive control for *Staphylococcus* strains, ampicillin for *L. monocytogenes*, chloramphenicol for *S. maltophilia* and gentamicin for *A. baumannii*.

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
