# Peer review of "Molecular Network and Culture Media Variation Reveal a Complex Metabolic Profile in Pantoea cf. eucrina D2 Associated with an Acidified Marine Sponge"

_ijms, 2020, doi:10.3390/ijms21176307_

Round 1

Reviewer 1 Report

Manuscript No. IJMS-907619

Title: Molecular Network and Culture Media Variation Reveal a Complex Metabolic Profile in Pantoea cf. eucrina D2 Associated to an Acidified Marine Sponge

The Authors isolated Pantoea from a marine sponge - Chondrosia reniformis. The strain was identified by 16S rRNA – Pantoea eucrina D2. The metabolome from the investigated strain was characterized by OSMAC (One Strain Many Compounds) [13] and Molecular Networking (MN). Specially attention was done to evaluate the diversity of secondary metabolites. The growth of Pantoea eucrina D2 and its production of metabolites were evaluated under different conditions, specially different inorganic nitrogen sources  (inorganic ammonium and nitrate)were added to the media culture.

The molecular networking analysis allowed the annotation of several metabolites that were clustered in three subgroups: lipoamino acids, diketopiperazines, surfactins. The production of specific metabolites was dependent by the composition of growth media.

It is the first report describes the production of surfactin derivatives, by a Pantoea strain.The major metabolites were isolated, chemically characterized by NMR and HR-ESI-MS analysis and subjected to antimicrobial assays towards human pathogens.

The Authors focused their attention on two latter classes: diketopiperazines, surfactins.

The article is written very good and the results presented are novel.

Final decision: The manuscript should be published in this form.

Author Response

Naples, Aug 27th, 2020

DearEditor,

Journal: International Journal of Molecular Sciences

Manuscript ID: ijms-907619

Special Issue: Microbial Biosurfactants, Current Research Trends and Applications

Title:Molecular Network and Culture Media Variation Reveal a Complex Metabolic Profile in Pantoea cf. eucrinaD2 Associated to an Acidified Marine Sponge

Authors:Giovanni Andrea Vitale, Martina Sciarretta, Chiara Cassiano, Carmine Buonocore, Carmen Festa, Valerio Mazzella, Laura Núñez Pons, Maria Valeria D’Auria, Donatella de Pascale

Thank you very much for the revision of the article 

The revised version contains all the constructive, up to point modifications requested by the Editor and Reviewer

Reviewer comments:

Reviewer 1

The manuscript contains interesting research material and deals with important issues. 

Point 1) It would be good to indicate in the introduction what is the essence of the article. Novelty and research hypothesis needs to be highlighted

Reply: Thank you for the suggestion, we added in “Track Changes” mode a brief part (lines 87, 89-94) in order to highlight novelties and outputs of this research.

Point 2) Please add NMR spectra to supplementary materials.

Reply: Thank you for the comment. In the supplementary file we reported mono and bidimensional experiments for the new compound 1 (1H, COSY, HSQC, HMBC), while for compounds 2-5 and 8 which have never been isolated from natural sources, but have been previously synthesized, 1H and 13C NMR are reported. Finally, for the known compounds 2, 6 and 7 we have reported only the 1H NMR spectrum.

For all the already mentioned compounds (natural and synthetic products) the data have been compared with literature, the appropriate citations are given in the main text.

Reviewer 3

Point 1) The work is very interesting. I have no major problems with the manuscript. What I do prior suggest to acceptance is a revision of the English. In some points is hard to understand given the bad sentences.

Reply: We revised the manuscript with the “Track Changes” function according to your comment. 

Now, we do hope that the manuscript is acceptable for publication.

Thank you in advance for your kind cooperation,

Yours sincerely

Reviewer 2 Report

The work is very interesting. I have no major problems with the manuscript. What I do prior suggest to acceptance is a revision of the English. In some points is hard to understand given the bad sentences.

Author Response

(The authors gave the same response as above.)

Reviewer 3 Report

The manuscript contains interesting research material and deals with important issues.
It would be good to indicate in the introduction what is the essence of the article. Novelty and research hypothesis needs to be highlighted.
Please add NMR spectra to suplementary materials.

Author Response

(The authors gave the same response as above.)
